# Photosynthesis without β-carotene

**Pengqi Xu[1], Volha U Chukhutsina[1], Wojciech J Nawrocki[1], Gert Schansker[1], Ludwik W Bielczynski[1], Yinghong Lu[2], Daniel Karcher[2], Ralph Bock[2], Roberta Croce[1]\***

[1]Biophysics of Photosynthesis, Department of Physics and Astronomy, Faculty of Sciences, VU University Amsterdam and LaserLab Amsterdam, Amsterdam, Netherlands; [2]Max Planck Institute of Molecular Plant Physiology, Potsdam-Golm, Germany

**Abstract** Carotenoids are essential in oxygenic photosynthesis: they stabilize the pigment–protein complexes, are active in harvesting sunlight and in photoprotection. In plants, they are present as carotenes and their oxygenated derivatives, xanthophylls. While mutant plants lacking xanthophylls are capable of photoautotrophic growth, no plants without carotenes in their photosystems have been reported so far, which has led to the common opinion that carotenes are essential for photosynthesis. Here, we report the first plant that grows photoautotrophically in the absence of carotenes: a tobacco plant containing only the xanthophyll astaxanthin. Surprisingly, both photosystems are fully functional despite their carotenoid-binding sites being occupied by astaxanthin instead of β-carotene or remaining empty (i.e. are not occupied by carotenoids). These plants display non-photochemical quenching, despite the absence of both zeaxanthin and lutein and show that tobacco can regulate the ratio between the two photosystems in a very large dynamic range to optimize electron transport.

## Introduction

Carotenoids form a large class of natural pigments responsible for the yellow, orange, and red colors of fruits and leaves (*Stange, 2016*). In the photosynthetic membranes, they are mainly associated with proteins, forming pigment–protein complexes. Their large absorption cross-section in the blue region of the solar spectrum makes them ideal light-harvesting pigments, especially for aquatic organisms (*Croce and van Amerongen, 2014*). However, the primary role of carotenoids in photosynthesis is photoprotection. Their capacity to quench chlorophyll (Chl) triplets (thus avoiding their reaction with molecular oxygen and the production of singlet oxygen), and to scavenge singlet oxygen make them essential for the survival of the organism (*Frank and Cogdell, 1996*; *Borth, 1975*; *Havaux, 1998*). In addition, carotenoids are involved in the quenching of singlet excited state Chls in a process known as non-photochemical quenching (NPQ), which controls the level of excited states in the membrane, thus protecting the photosynthetic apparatus from high light damage (*Ruban et al., 2012*).

Two species of carotenoids are present in the photosynthetic membranes: carotenes and their oxygenated derivatives, xanthophylls. The main carotene, β-carotene (β-car), is associated with the core of photosystems I and II (*Umena et al., 2011*; *Qin et al., 2015*), and is present in all organisms performing oxygenic photosynthesis. The xanthophylls (in plants mainly lutein (Lut), neoxanthin (Neo), violaxanthin (Vio) and zeaxanthin (Zea)), instead, are bound to the light-harvesting complexes (LHCs) that act as peripheral antennae increasing the absorption cross-section of both photosystems (*Qin et al., 2015*; *Su et al., 2017*). LHCs are able to accommodate different xanthophylls, but they cannot fold in the presence of β-carotene only (*Croce et al., 1999*; *Phillip et al., 2002*). Also, PSII assembly has been suggested to require the presence of carotenes (*Masamoto et al., 2004*), while PSI is stable also in the absence of carotenoids (*Masamoto et al., 2004*; *Santabarbara et al., 2013*).

**\*For correspondence:**
r.croce@vu.nl

**Competing interests:** The authors declare that no competing interests exist.

**eLife digest** Most life on Earth depends on photosynthesis, the process used by plants and many other organisms to store energy from sunlight and produce oxygen. The first steps of photosynthesis, the capture and conversion of sunlight into chemical energy, happen in large assemblies of proteins containing many pigment molecules called photosystems. In plants, the pigments involved in photosynthesis are green chlorophylls and carotenoids. In addition to harvesting light, carotenoids have an important role in preventing damage caused by overexposure to sunlight

There are over one thousand different carotenoids in living beings, but only one, β-carotene, is present in every organism that performs the type of photosynthesis in which oxygen is released, and is thought to be essential for the process. However, this could never be proved because it is impossible to remove β-carotene from cells using typical genetic approaches without affecting all other carotenoids.

Xu et al. used genetic engineering to create tobacco plants that produced a pigment called astaxanthin in place of β-carotene. Astaxanthin is a carotenoid from salmon and shrimp, not normally found in plants. These plants are the first living things known to perform photosynthesis without β-carotene and demonstrate that this pigment is not essential for photosynthesis as long as other carotenoids are present. Xu et al. also show that the photosystems can adapt to using different carotenoids, and can even operate with a reduced number of them.

Xu et al's findings show the high flexibility of photosynthesis in plants, which are able to incorporate non-native elements to the process. These results are also important in the context of increasing the photosynthetic efficiency, and thus the productivity of crops, since they show that a radical redesign of the photosynthetic machinery is feasible.

While mutants lacking individual or all xanthophylls but still containing carotenes have been identified for several organisms (e.g. *Dall'Osto et al., 2013*; *Ware et al., 2016*; *Niyogi et al., 1997*; *Pogson et al., 1998*; *Domonkos et al., 2013*; *Schäfer et al., 2005*), mutants lacking carotenes have only been isolated in cyanobacteria and the green alga *Chlamydomonas reinhardtii* when these organisms were grown heterotrophically (*Santabarbara et al., 2013*; *Sozer et al., 2010*; *Tóth et al., 2015*). In these mutants, no PSII was assembled. This finding, together with the fact that no PSII complexes without carotenes have ever been observed, have suggested that carotenes have a vital role not only in photosynthesis but also for the survival of the plant cell (*Dall'Osto et al., 2014*). However, this assumption could never be verified, because available mutants without carotenes completely lack carotenoids.

In this work, we have analyzed tobacco (*Nicotiana tabacum*) plants in which the carotenoid biosynthetic pathway was engineered (by stable transformation of the chloroplast genome) to only produce the ketocarotenoid astaxanthin (*Lu et al., 2017*; *Figure 1*). The physiological characteristics and the autotrophic growth of these plants demonstrate that photosynthesis without carotenes is possible, at least when plants are grown in laboratory conditions.

## Results and discussion

The leaves of the tobacco mutant are orange at an early stage and become greener with age (see *Figure 1*, *Table 1*). This is likely due to the high expression of the chloroplast genome in young leaves (*Edwards et al., 2010*), which in the mutant results in the massive production of astaxanthin. Most of the astaxanthin is present in the form of crystals or aggregates in the chloroplast (*Lu et al., 2017*). The high-level synthesis of astaxanthin uses a substantial part of the plant's energy budget and fixed carbon and may contribute to the slow growth of the plants. This negative effect on growth is likely exacerbated by the fact that astaxanthin absorbs most of the incident light, decreasing the number of photons available for photosynthesis. Indeed, the greening of the leaves corresponds to an increase in the growth rate of the plants. Both young and mature leaves of the mutant plants contain only 20% of the Chls per fresh weight as compared to the wild-type (WT), but have a similar (mature leaves) or far higher (young leaves) carotenoid content (*Table 1*). However, the mutant, at all stages of growth, only contains astaxanthin and traces of by-products of astaxanthin

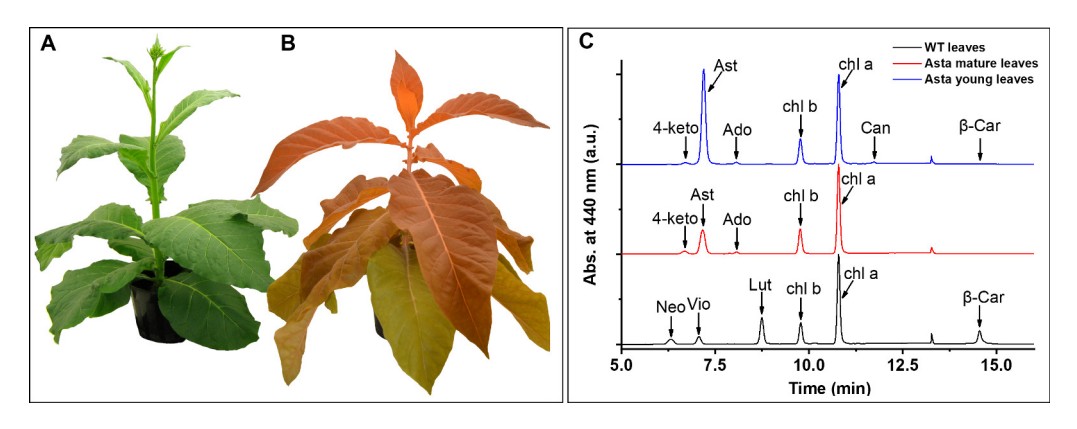

**Figure 1.** Phenotype of wild-type (WT) and astaxanthin-synthesizing tobacco plants (Asta) and pigment analysis. WT (**A**) and Asta (**B**) plants, 7 and 21 weeks old, respectively. Note that the older leaves in the mutant are less orange than the young leaves. See also *Figure 1—figure supplement 1*. (**C**) Chromatographic profiles of the pigments extracted from leaves normalized to the Chl a peak. Neo, neoxanthin; Vio, violaxanthin; Lut, lutein; β-car, β-carotene; Chl, chlorophyll; 4-keto, 4-ketoanteraxanthin; Ast, astaxanthin; Ado, adonixanthin; Can, Canthaxanthin.

The online version of this article includes the following figure supplement(s) for figure 1:

**Figure supplement 1.** Plants at different ages.

synthesis (*Hasunuma et al., 2008*) and does not accumulate (<0.005 times the WT) the carotenoids that are usually present in the WT (*Figure 1*). This result is different from the analysis of previously generated astaxanthin-producing plants that still contained WT carotenoids, although in reduced amounts (*Hasunuma et al., 2008*; *Fujii et al., 2016*; *Röding et al., 2015*). Thus, our engineered tobacco (hereafter referred to as Asta) represents the first organism showing autotrophic growth in the virtual absence of carotenes. In the following, we report the experiments performed on mature leaves, which have a Chl/car similar to the WT.

Since violaxanthin and lutein are considered to be necessary for the folding of the antenna complexes (*Dall'Osto et al., 2006*), and β-carotene was thought to be required for PSII assembly and photosynthetic activity (*Santabarbara et al., 2013*; *Sozer et al., 2010*; *Tóth et al., 2015*), we analyzed the effect of their absence on the composition and organization of the photosynthetic apparatus. 2D gel electrophoresis (*Figure 2*) and immunoblot analyses (*Figure 2—figure supplement 1*) of thylakoid membranes show that all of the main photosynthetic proteins are present in Asta plants, but the PSII/PSI ratio is far higher than in the WT (*Figure 2—figure supplement 1*). The LHC/PSII ratio is, however, similar, except for the antenna protein Lhcb5 that is strongly reduced, and for PsbS, the main protein involved in NPQ (*Li et al., 2000*), which is increased in the mutant (*Figure 2—figure supplement 1*). PSI-LHCI, ATP synthase and cytochrome $b_6f$ have the same mobility in native gels as the WT complexes, indicating that they are stable and have the same supramolecular organization. By contrast, the stability of PSII seems to be affected as the bands corresponding to PSII supercomplexes and LHCII trimers, which are well defined in the WT, are substituted by a smear in the mutant, suggesting that the PSII complexes are more heterogeneous, incompletely assembled or less stable than in the WT (*Figure 2*).

**Table 1.** Pigment composition of leaves.

| Samples | Chl a/b | Chl/car | Chl/fresh wt (mg/g) | Chl/leaf area # (mg) |
|---|---|---|---|---|
| WT | 3.79 ± 0.09 | 4.21 ± 0.20 | 2.63 ± 0.39 | 0.0312 ± 0.0026 |
| Asta mature leaves | 3.01 ± 0.10 | 3.02 ± 0.22 | 0.58 ± 0.07 | 0.0062 ± 0.0008 |
| Asta young leaves | 3.16 ± 0.19 | 0.76 ± 0.13 | 0.55 ± 0.09 | 0.0049 ± 0.0011 |

(#50 mm$^2$; Average values ± SD are shown. n = 10 biological replicas).

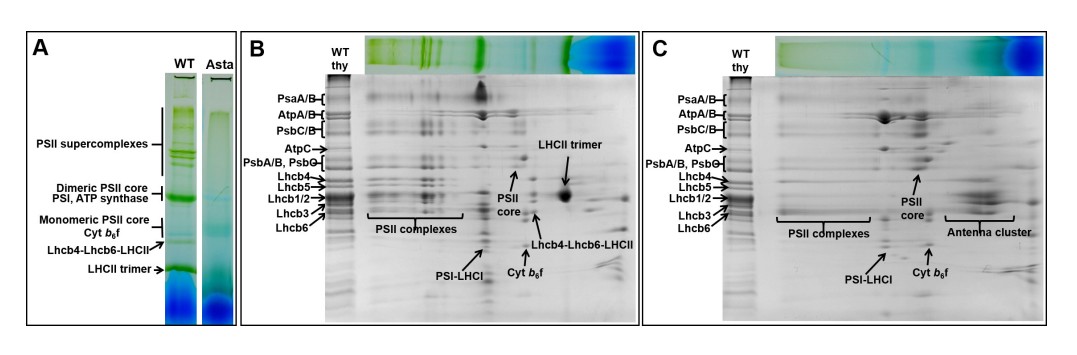

**Figure 2.** Protein composition and supramolecular organization of the photosynthetic complexes in WT and Asta plants. (A) Thylakoids were solubilized with 1% α-DDM and loaded on a blue-native gel. Second dimension SDS-PAGE of the WT (B) and the Asta mutant (C). The immunoblotting analysis is shown in *Figure 2—figure supplement 1*.

The online version of this article includes the following figure supplement(s) for figure 2:

**Figure supplement 1.** Comparison of the protein composition of WT and Asta plants.

Next, we investigated the effects of the change in carotenoid composition on the properties of the individual complexes that were isolated from thylakoid membranes and separated by sucrose density gradient ultracentrifugation (*Figure 3—figure supplement 1*). Pigment analysis (*Table 2* and *Figure 3—figure supplement 2*) confirmed that astaxanthin is the only carotenoid associated with all pigment-binding complexes in Asta plants, while β-carotene is present in PSI in a highly substoichiometric amount (0.15 β-carotene molecules per complex) and it is virtually absent in PSII (0.03 molecules per complex). This means that most of the PSI and PSII complexes in the mutant plants do not contain β-carotene at all. Normalized to Chl, Asta-LHCs contain the same number of carotenoids as the WT monomers, but instead of binding lutein, neoxanthin and violaxanthin, they only bind astaxanthin, indicating that all carotenoid-binding sites are promiscuous and can accommodate different xanthophylls. The pigment analysis also showed that β-carotene can be substituted by astaxanthin in both PSII and PSI cores. However, the higher Chl/car ratio in the isolated Asta complexes compared to the WT complexes indicates that not all sites that are occupied by β-carotene in the WT are occupied by astaxanthin in the mutant complexes, but some are left 'empty' in that they are not occupied by carotenoids. Although we cannot exclude the possibility that some of the astaxanthin molecules are more loosely bound and thus are lost during purification, the fact that both PSI and PSII complexes can be purified with a large number of 'empty' sites indicates that their occupancy by carotenoids is not crucial for the stability of the complexes.

Interestingly, absorption (*Figure 3—figure supplement 3A*) and circular dichroism (*Figure 3—figure supplement 3B*) spectra of LHCs and PSII core complexes from the WT and the mutant are very similar (see *Figure 3—figure supplement 3* for a more detailed explanation) in the Chl

**Table 2.** Pigment composition of isolated thylakoidal complexes.

| Samples | Chl a/b | Chl/car | Lute+neo+viola | β-caro | Asta and its bypass products | Total chls* |
|---|---|---|---|---|---|---|
| WT Lhcb Monomers | 2.41 ± 0.01 | 4.1 ± 0.03 | 2.86 ± 0.01 | 0.06 ± 0.01 | n.d. | 12 |
| WT LHCII trimer | 1.40 ± 0.00 | 3.60 ± 0.01 | 3.90 ± 0.01 | n.d. | n.d. | 14 |
| Asta-Lhcb mon | 1.48 ± 0.01 | 4.50 ± 0.03 | n.d. | n.d. | 3.1 ± 0.02 | 14 |
| WT PSII | 8.88 ± 0.12 | 5.00 ± 0.02 | 3.47 ± 0.08 | 3.93 ± 0.08 | n.d. | 37 |
| Asta PSII | 7.74 ± 0.27 | 8.90 ± 0.17 | n.d. | 0.03 ± 0.01 | 4.12 ± 0.01 | 37 |
| WT PSI-LHCI | 9.29 ± 0.12 | 4.60 ± 0.02 | 14.37 ± 0.16 | 19.56 ± 0.16 | n.d. | 156 |
| Asta-PSI-LHCI | 5.44 ± 0.19 | 8.10 ± 0.18 | n.d. | 0.15 ± 0.03 | 19.11 ± 0.03 | 156 |

*Total Chls are based on values reported in the literature for the WT complexes (*Qin et al., 2015*; *Su et al., 2017*). The chromatograms are shown in *Figure 3—figure supplement 2*. (Average values ± SD are shown. n ≥ 3 biological replicas, n.d. = not detected).

absorption regions. This indicates that there are no significant changes in the pigment organization of the complexes and thus in their three-dimensional structure. The only exception is Asta-PSI-LHCI, the fluorescence emission of which showed a 6 nm shift to shorter wavelengths as compared to the WT complex (*Figure 3—figure supplement 3C* and *Figure 3—figure supplement 4*). Since the PSI emission at 77 K originates mostly from two specific Chls (called far-red Chls) of Lhca3 and Lhca4 (*Morosinotto et al., 2003*), we can conclude that the interaction between these Chls is slightly changed in the mutant.

Carotenoids are known to be required for the stability of the pigment-binding holoproteins (*Paulsen et al., 1993*). Our data measured on the isolated complexes show that the difference in composition between WT and mutant complexes influences the denaturation temperature by only 5–10°C (*Figure 3A*). This is surprising considering that several of the carotenoid-binding sites in the isolated PSI and PSII are not occupied by carotenoids and indicates that only some of them play a crucial role in protein stability.

Photoprotection via Chl triplet quenching and singlet oxygen scavenging is the primary role of carotenoids in photosynthesis (*Siefermann-Harms, 1987*). Photobleaching experiments (*Figure 3B*) show that, while the photostability of LHCs and PSII core is only partially affected by the change in carotenoid composition, Asta-PSI-LHCI is far more sensitive to light than the WT complex. It is likely that this effect on PSI is due to the reduced number of carotenoid molecules associated with the complex, which results in less efficient Chl triplet quenching. However, it is worth noting that, even with a large part of the carotenoid-binding sites not occupied by carotenoids, PSI is more photostable than PSII-WT, in agreement with the fact that, in PSII, carotenoids cannot provide protection by

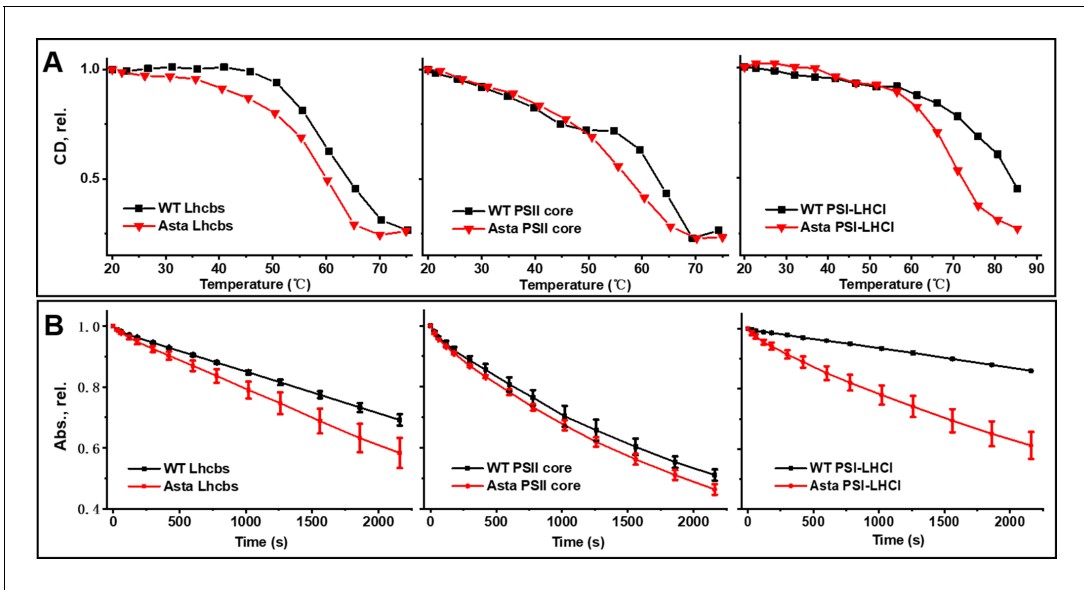

**Figure 3.** Thermal and photo-stability of photosynthetic complexes. (**A**) Thermal denaturation was monitored by following the CD signals in the $Q_y$ (Lhcb: 610–700 nm, PSII core: 640–700 nm, PSI-LHCI: 675–735 nm) region at increasing temperature. (**B**) Photobleaching is measured as the decrease in absorption of the area in the $Q_y$ region (600–750 nm) as a function of the length of the high light treatment (Average values ± SD are shown for n = 3 technical replicas.). The purification of the complexes is shown in *Figure 3—figure supplement 1*. The pigment analysis of the purified complexes is shown in *Figure 3—figure supplement 2*. The absorption, CD, and fluorescence emission spectra of the complexes are shown in *Figure 3—figure supplement 3* for comparison the fluorescence spectra of the thylakoids are shown in *Figure 3—figure supplement 4*. The raw data used for this figure are provided in *Figure 3—figure supplement 5*.

The online version of this article includes the following figure supplement(s) for figure 3:

**Figure supplement 1.** Isolation of photosynthetic complexes.

**Figure supplement 2.** Pigment analysis.

**Figure supplement 3.** Absorption, CD, and 77K fluorescence spectra of isolated complexes from WT (black) and Asta (red).

**Figure supplement 4.** Fluorescence emission spectra of thylakoids at 77 K.

**Figure supplement 5.** Original CD (**A**) and absorption (**B**) spectra.

quenching singlet oxygen formed via P680 triplet because of the very high oxidizing potential of PSII (*Telfer, 2014*).

Next, we investigated the effect of the substitution of carotenes with astaxanthin on the light-harvesting and trapping properties of the photosynthetic complexes in vivo by performing time-resolved fluorescence measurements at 20°C on intact leaves (*Figure 4* and *Figure 4—figure supplement 1*). The PSI kinetics is very similar in the WT (70 ps) and in the mutant (65 ps), and the small difference can be ascribed to the reduced far-red Chl content of Asta-PSI-LHCI, which is known to influence the PSI trapping time (*Croce and van Amerongen, 2013*). The PSII kinetics in the mutant leaves changes in the presence versus absence of photochemistry (measurements performed with the reaction center (RC) open and closed, respectively) as it does in the WT, indicating that excitation energy transfer occurs in the mutant and the harvested energy is used for photochemistry. However, all the kinetics are faster and the difference between closed and open RC is smaller in mutant than in WT leaves, suggesting that the antenna complexes of the mutant plants are statically quenched in vivo. Measurements on isolated Asta-Lhcb show that this is indeed the case: these complexes are strongly quenched (lifetime of 0.87 ns vs. 3.5 ns in the WT; *Figure 4—figure supplement 2*) due to the presence of astaxanthin (*Liguori et al., 2017*). It has also been shown that part of the astaxanthin population can transfer excitation energy to the chlorophylls, thus also acting as light-harvesting pigment (*Liguori et al., 2017*).

The presence of excitation energy transfer from the antenna to the RC in mutant leaves indicates that, although the interactions between the building blocks of the PSII supercomplex are not strong enough to survive purification (see *Figure 2*), the supercomplexes are functional in vivo meaning that, in the membrane, the subunits are close enough to each other to ensure the delivery of the harvested energy to the reaction center. Indeed, the short excited state lifetime of the antenna (indicative of a constitutively quenched antenna) in the mutant can fully account for the lower maximum quantum efficiency of PSII ($F_V/F_M$; *Table 3*) in mutant plants, which is mainly the result of low fluorescence emission in the absence of photochemistry ($F_M$).

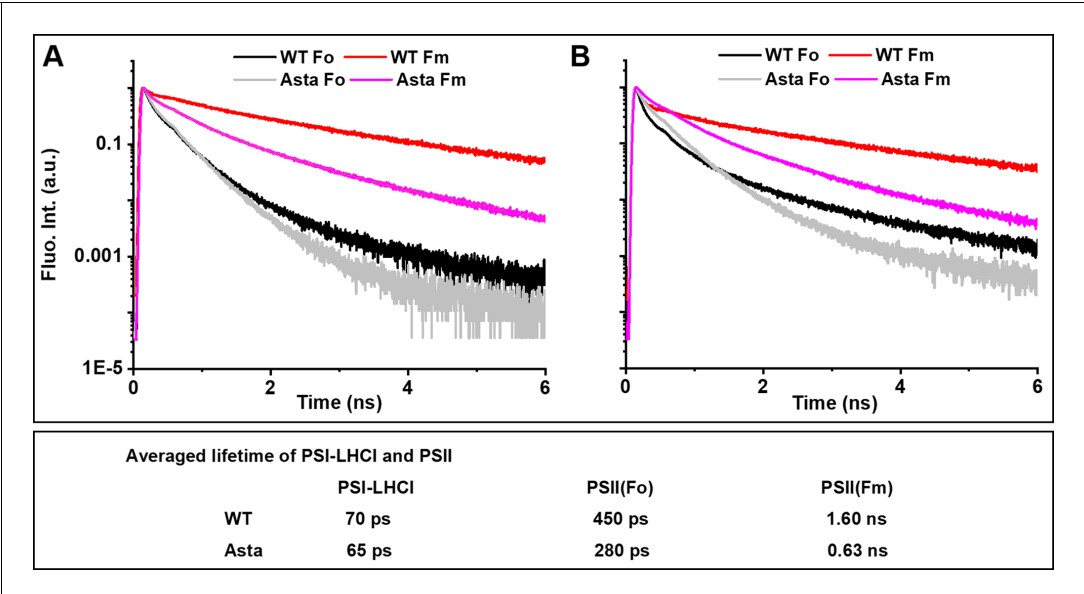

**Figure 4.** Normalized fluorescence decay traces measured at room temperature on WT and Asta leaves in the presence ($F_0$) or absence ($F_M$) of photochemistry. Fluorescence was detected at (A) λ = 685 nm and (B) λ = 720 nm. respectively. Note that, although the decay at 680 nm is dominated by PSII and at 720 nm by PSI, both complexes contribute to the decay at both wavelengths (see *Figure 4—figure supplement 1* for the analysis of the complete data set with spectral resolution). The fluorescence decay and analysis of the purified LHCII are shown in *Figure 4—figure supplement 2*. The online version of this article includes the following figure supplement(s) for figure 4:

**Figure supplement 1.** Results of time-resolved fluorescence of leaves from WT (**A and B**) and Asta (**C and D**).

**Figure supplement 2.** Time-resolved fluorescence decay traces of Lhcb.

Finally, we analyzed the photosynthetic performance of the Asta plants. Electrochromic shift (ECS) of the carotenoid absorption is commonly used to study the function of all major photosynthetic complexes (*Bailleul et al., 2015*). We verified that the mutant plants exhibit an ECS signal and we determined its light-induced difference spectrum, which agreed with the prediction that astaxanthin is solely responsible for this in vivo Stark effect (*Figure 5—figure supplement 1*). Using ECS, we observed that the functional PSII/PSI RC ratio was far larger in the mutant than in the WT, in qualitative agreement with our biochemical data (*Table 3*; *Figure 2—figure supplement 1*). The difference in the values obtained with the two methods is partially due to the limited quantitative power of immunoblots, but also suggests that some of the PSII cores are not functional. The high PSII/PSI ratio in the mutant seems to be a compensation mechanism for the decrease in the relative functional PSII/PSI antenna size (measured with two independent methods; *Table 3* and *Figure 5—figure supplement 2*) observed in the Asta plants, which is due to the presence of static quenching.

Indeed, comparison of the steady-state photochemical yields of PSII and PSI revealed that, at all light intensities, in both WT and mutant plants, the balance between PSII and PSI photochemistry is maintained (*Figure 5A* and *Figure 5—figure supplements 3* and *4*), meaning that the plants are able to compensate for the strong decrease in the PSII functional antenna size by decreasing the PSI/PSII ratio. This means that these plants have the capacity to modulate the PSI/PSII ratio in a large dynamic range. Finally, transient $Q_A$ reduction and reoxidation kinetics suggest that no significant differences in the PSII electron transfer occur in the mutant plants (*Figure 5—figure supplements 2* and *5*).

The full operational capacity of the electron transport chain permitted us to verify whether the photoprotective regulation is maintained in the mutant plants. As expected, the NPQ amplitude was largely reduced in the mutant (*Figure 5B* and *Figure 5—figure supplement 6*), because the ΔpH-induced, PsbS-dependent quenching has to compete with the strong, constitutive astaxanthin quenching in these plants. Note that the difference in NPQ level (1.8 in the WT vs. 0.3 in the mutant) can be fully ascribed to the presence of the static quencher in the mutant, which strongly reduces the maximal fluorescence in both dark ($F_M$) and light ($F_M'$) states. This is supported by the $NPQ_{(t)}$ calculation, which permits to correct the apparent NPQ for the presence of a pre-existing quenching, assuming that a decrease of the $F_V/F_M$ value is solely due to this static quenching. The data show that NPQ(t) is even larger in the mutant than in the WT (*Figure 5—figure supplement 7*).

Importantly, despite the difference in apparent NPQ amplitude, the kinetics of onset and recovery are identical to those of the WT (inset in *Figure 5B*) and consistent with $q_E$ characteristics. This outcome is particularly striking if one considers that Asta plants lack both lutein and zeaxanthin, which are believed to be essential for NPQ (*Niyogi et al., 1998*). It is likely that the high amount of PsbS in the mutant (*Figure 2—figure supplement 1*) can compensate for the lack of the xanthophyll cycle, or that astaxanthin can also be responsible for the dynamic quenching. Whatever the reason for the presence of NPQ in the mutant, our Asta plants clearly show that lutein and zeaxanthin are not absolutely necessary for it.

In summary, we have shown that the carotenoid-binding sites of the core complexes of PSI and PSII are promiscuous. Although they bind carotenes in all known photosynthetic organisms, our data demonstrate that they can also accommodate xanthophylls. This is at variance with the LHCs that can bind various xanthophylls but cannot fold with carotenes. More importantly, we show that both PSI and PSII are stable while most of their carotenoid-binding sites are not occupied by carotenoids and the rest is occupied by an alien xanthophyll. These results indicate that the core complexes are even more robust than the outer antennae and can endure radical changes even in one of their main components. In this respect, it is important to realize that the difference in growth rate between WT and mutant plants is not due to the absence of carotenes, but rather to the presence of astaxanthin that stabilizes the LHCs in a quenched conformation. In conclusion, the substitution of carotenes

**Table 3.** In vivo photosystem II parameters.

| | $F_V/F_M$* | PSII:PSI ratio (ECS)# | PSII:PSI antenna size (ECS)# | Relative PSII antenna size (fluorescence)§§ |
|---|---|---|---|---|
| WT | 0.82 ± 0.01 | 1.09 ± 0.12 | 1 ± 0.59 | 1 ± 0.04 |
| Asta | 0.43 ± 0.03 | 2.6 ± 0.33 | 0.33 ± 0.05 | 0.33 ± 0.02 |

(*Average values ± SD are shown. *n = 15, #n = 4, §§n = 3 leaves/plants*).

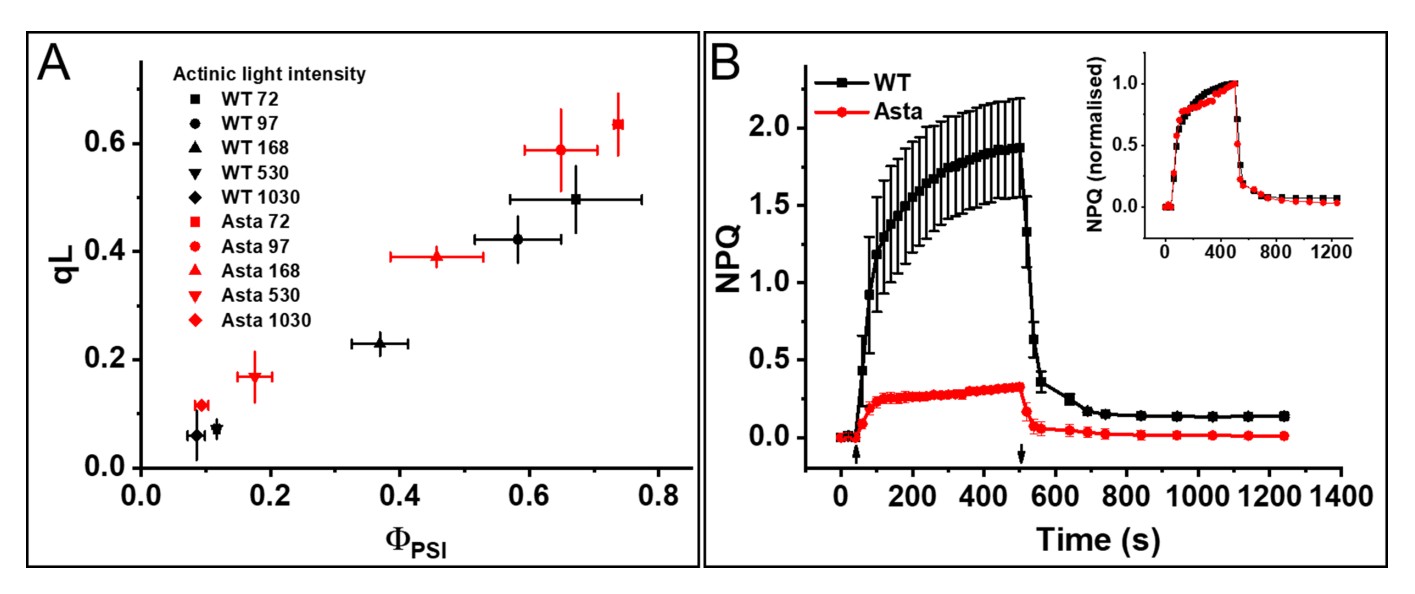

**Figure 5.** Photosynthetic performance and photoprotection. (**A**) Relation between qL and $\Phi_{PSI}$ measured at various light intensities (70–1030 µmol photons m$^{-2}$ s$^{-1}$). (**B**) NPQ kinetics upon transition from the dark-to-light (upward arrow,) and subsequent relaxation in the dark (downward arrow). Average values ± SD are shown for n = 3 leaves/plants. The inset shows amplitude-normalized kinetics. See also *Figure 5—figure supplements 1–7*. The online version of this article includes the following figure supplement(s) for figure 5:

**Figure supplement 1.** Electrochromic shift: absorption changes induced by the formation of a trans-thylakoid electric field (Δψ) in vivo in Asta plants.
**Figure supplement 2.** Light-intensity dependence (µE m$^{-2}$ s$^{-1}$ red light) of the OJIP transients of three types of tobacco leaves.
**Figure supplement 3.** qP (A) and qL (B) as a function of light intensity in WT and Asta leaves.
**Figure supplement 4.** Relation between photochemical quenching (qP) and $\Phi_{PSI}$ measured at various light intensities (70–1030 µmol photons m$^{-2}$ s$^{-1}$).
**Figure supplement 5.** $Q_A$- reoxidation kinetics.
**Figure supplement 6.** Light dependence of NPQ.
**Figure supplement 7.** NPQ(t) in WT and Asta mutant.

with the xanthophyll astaxanthin does not impair the functional assembly of the photosynthetic apparatus, nor does it impede efficient electron transfer and NPQ, demonstrating that carotenes are not essential neither for the biosynthesis of the photosynthetic apparatus nor for its function. This finding has important implications not only for our understanding of the structure and function of the photosynthetic apparatus but also for future efforts to design synthetic photosystems with novel and improved properties.

## Materials and methods

### Tobacco growth and thylakoid isolation

Seeds from mutant and WT plants were sown on moist filter paper and synchronized at 4°C for 2–3 days before being moved to room temperature until germination. The seedlings were transferred to soil and grown at 22°C under 150–200 µmol photons m$^{-2}$ s$^{-1}$ for the WT and 80–120 µmol photons m$^{-2}$ s$^{-1}$ for the mutant with 14 hr of light per day. Plants were fed with commercial fertilizer each week. Leaves from WT (5–6 weeks old) and Asta plants (around 20 weeks for younger leaves, 24–30 weeks for older leaves) were used for physiological measurements and thylakoid isolation. WT thylakoid isolation was performed as described in *Xu et al., 2015*. The centrifuge speed was increased to 4000 g for the first step of the isolation from mutant tobacco.

### Pigment analysis

Pigments from isolated protein–pigment complexes or leaves were extracted with 80% acetone. HPLC was performed as in *Xu et al., 2015* with the modification that buffer B was linearly increased from 0 to 100% in 9.2 min. Chlorophyll *a/b* ratios and chlorophyll/carotenoid ratios were calculated

by fitting their individual absorption spectra to measured spectra (*Xu et al., 2015*). Examples of the fitting of total thylakoids and isolated Lhcbs are shown in *Figure 3—figure supplement 2* panels C and D.

## Blue-native gel electrophoresis, SDS-PAGE, immunoblotting, and sucrose density gradient centrifugation

Blue-native gels were performed as described in *Järvi et al., 2011* with the modifications described in *Bielczynski et al., 2016*. The second dimension and the SDS-PAGE were performed as described in *Schägger, 2006*.

For immunoblot analysis, total protein extracts were separated by SDS-PAGE and transferred to a Protran 0.45 mm nitrocellulose membrane. Specific primary antibodies (Agrisera) were used to detect the target proteins. Chemiluminescence was detected using an ImageQuant LAS 4000 imaging system.

For sucrose density gradient fractionation, thylakoids equivalent to 0.2 mg total chlorophyll were washed with 5 mM EDTA and resuspended in 200 µL 10 mM Hepes (pH 7.5). An equal volume of 1.2% α-DDM was added, mixed gently, and the solubilized thylakoids were centrifuged at 14,000 rpm for 10 min at 4℃. The supernatant was loaded on a 0–1 M sucrose gradient (10 mM Hepes, pH 7.5, 0.03% α-DDM) and centrifuged at 288,000 g for 17 hr. The separated bands were collected with a syringe.

## State-steady spectroscopy measurements

### Absorption and CD

Absorption spectra were measured at room temperature with a Varian Cary 4000 UV-Vis-spectrophotometer. CD spectra were recorded using a Chirascan-Plus spectropolarimeter (Applied Photophysics) at 20℃. The OD of the samples was 0.8–1/cm at the maximum of the Qy region.

### 77 K fluorescence emission

Low-temperature fluorescence emission spectra were recorded using a Fluorolog 3.22 spectrofluorometer (Jobin Yvon-Spex). For 77 K measurements, a home-built liquid nitrogen-cooled device was used. The samples were excited at 440 nm and the fluorescence emission was detected in the 600–800 nm range. Excitation and emission slit widths were set to 3 nm.

All measurements were performed in the same buffers used for the sucrose gradients.

## Photobleaching and protein stability

### Photobleaching

The samples were diluted to an absorbance of around 0.8 at the maximum in the Qy region. The protein–pigment complexes were illuminated with white light (7100 µmol photons $m^{-2}$ $s^{-1}$) from a halogen lamp with optic fiber arm. After each interval, the cuvette was removed from the light beam, and the absorption spectra were recorded with a Varian Cary 4000 UV-Vis-spectrophotometer in the range between 600 and 750 nm (*Croce et al., 1999*).

### Protein stability

The stability of the isolated complexes was tested by measuring the temperature denaturation curve as obtained by monitoring changes in the CD spectra in the Qy region while increasing the temperature from 20 to 90℃ (*Croce et al., 1999*). A 400 µL sample with OD 0.8 at the maximum in the Qy region was used in this measurement.

## In vivo time-resolved fluorescence measurements

Time-resolved fluorescence measurements on leaves were done using a time-correlated single photon-counting (TCSPC) setup as described previously (*Chukhutsina et al., 2019*). Excitation at 650 nm was used to excite Chl *b* preferentially. Detached plant leaves were placed between two glass plates and mounted in the rotation cuvette (diameter: 10 cm; thickness: 1 mm). The cuvette was rotated at 1400 rpm while oscillating sideways. Fluorescence was measured in a front-face arrangement from the upper side of the leaves. Time-resolved fluorescence decays were measured at multiple detection wavelengths (between 675 and 690 nm with a wavelength step of 5 nm, and between

700 and 760 nm with a maximal wavelength step of 10 nm). The measurements were done in the presence and in the absence of PSII photochemistry (open ($F_0$) and closed ($F_M$) states, respectively).

i. $F_0$ was measured in complete darkness after overnight dark adaptation. The repetition rate was then reduced by a Pulse Picker (Spectra Physics) from 40 to 0.8 MHz. The excitation power was 20 µW. Preliminary checks with different powers and repetition rates were done to ensure that the PSII reaction centers (RCs) remained indeed open during the measurement.

ii. To measure leaves with closed PSII RCs ($F_M$), the leaves were incubated for 12 hr in sucrose (0.3 M) with addition of 50 µM 3-(3,4-dichlorophenyl)−1,1-dimethylurea (DCMU). To achieve full closure of the PSII RCs during the measurement, additional blue LED light of low intensity (~50 µmol photons·m$^{-2}$·s$^{-1}$) was used to preilluminate leaves just before detection of the signal. The repetition rate was reduced by a Pulse Picker from 40 MHz to 4 MHz. The excitation power was 100 µW.

The measurement time at a single wavelength was limited to 10 min, to avoid changes in the leaves due to prolonged measurement in the rotating cuvette. To perform an experiment in one state took 2–3 hr. All in vivo measurements were performed at 20℃. The obtained fluorescence decay traces were analyzed globally with the 'TRFA Data Processing Package' of the Scientific Software Technologies Center (Belarusian State University, Minsk, Belarus) (*Digris et al., 1999*). The global analysis methodology is described in *van Stokkum et al., 2004*. In short, a number of parallel, non-interacting kinetic components was used as a kinetic model, so the total dataset was fitted with function $f(t, \lambda)$ as follows:

$$\sum_{1,2\ldots}^{N} DAS_i(\lambda)\exp(-\frac{t}{\tau_i}) \oplus irf(t,\lambda),$$

where the decay-associated spectrum ($DAS_i$) is the amplitude factor associated with a decay component $i$ having a decay lifetime $\tau_i$, and $irf(t, \lambda)$ was measured using scattering light. Typical full-width at half-maximum (FWHM) values were 28 ± 2 ps.

## In vitro time-resolved fluorescence measurement

Time-resolved fluorescence measurements on isolated LHCII were performed on a FluoTime200 setup (Picoquant). The samples were diluted to an OD of 0.05 cm$^{-1}$ at the maximum in the Qy region and measured in a 3.5 mL cuvette with a path length of 1 cm at 283 K. Excitation was provided by a 468 nm laser diode (preferential Chl *b* excitation) operating at 10 MHz repetition rate. The instrument response function (IRF) was obtained by measuring the decay of a pinacyanol iodide dye dissolved in methanol, which has a six ps fluorescence lifetime (*van Oort et al., 2008*). The resulting IRF FWHM was ~88 ps. The fluorescence decay kinetics was detected at 680 nm with a channel time spacing of 8 ps. Data analysis was performed by the TRFA DATA software as described above.

## ECS-based measurements

The ECS light-induced difference spectrum was determined according to *Bailleul et al., 2015* using a JTS-10 spectrophotometer (BioLogic, Grenoble, France). In brief, the leaf was subjected to a saturating pulse of red light (3000 µmol photons m$^{-2}$ s$^{-1}$; 80 ms), and the absorption changes at each wavelength after the pulse were recorded without additional actinic light. The baseline obtained without the saturating pulse was subtracted, and the values between 100 and 200 ms after the pulse (to avoid the contribution of signals due to rapid redox-changes of cytochromes) were averaged. The obtained spectrum closely matches the theoretical ECS spectrum of pure astaxanthin, which is [1-($\frac{df}{dx}$)] of astaxanthin-detergent solution spectrum (*Bailleul et al., 2010*).

The PSII:PSI RC ratio was determined using the JTS-10 spectrophotometer using saturating single-turnover laser flashes (five ns duration) provided by a dye laser pumped with a Nd:YAG laser (Minilite, Continuum) using the protocol described in *Nawrocki et al., 2016* but adapted for leaves. For the PSII+PSI signal, the leaf was infiltrated with water, and to obtain a pure PSI signal the leaf was infiltrated with hydroxylamine (HA, 1 mM) and 3-(3,4-dichlorophenyl)−1,1-dimethylurea (DCMU, 10 µM; both from Sigma), after a systematic verification that no variable fluorescence, and thus PSII

activity, remained in the leaf. ECS was detected at 546 nm (Asta) and 520–546 nm (WT) using weak white light LED pulses filtered with a 10 nm FWHM interference filter. The peak amplitude at 546 nm allows the detection at the isosbestic point of cyt. $b_6f$ haems (*Alric et al., 2005*). The functional antenna size was measured as described in *Nawrocki et al., 2016* but with 300 µmol photos m$^{-2}$ s$^{-1}$ red actinic light (630 nm peak) and detecting light as described above. The quantities of active PSII were corrected for the ~20% slowly-opening RCs accumulating after actinic light.

## Photosynthetic parameter measurements

### PSI and PSII redox state

Dark-adapted plants were measured with a Dual-PAM-100 (Walz) to record $q_L$ and P700 at different actinic light intensities (70–1030 µmol photons·m$^{-2}$·s$^{-1}$). Leaves dark-adapted overnight were illuminated for 10 min to attain a steady state. The 820 nm absorbance signal corrected for absorbance changes at 870 nm was used for the analysis of the P700 kinetics. $q_L$ was calculated according to *Kramer et al., 2004*.

### NPQ

Dark-adapted plants were measured by a Dual-PAM-100 (Walz) with a modulated measuring light of 7 µmol photons m$^{-2}$ s$^{-1}$ to keep the reaction centers in the open state, and a 4000 µmol photons m$^{-2}$ s$^{-1}$ (500 ms) saturating pulse to close the reaction centers. Actinic light of 531 µmol photons m$^{-2}$ s$^{-1}$ was used to induce NPQ. $NPQ_{(T)}$ was calculated as described in *Tietz et al., 2017*.

### OJIP

A HandyPEA (Hansatech Instruments Ltd, UK) was used to measure fluorescence induction upon a dark-to-light transition. Three red LEDs (peak intensity at ~650 nm) were used as a light source, giving approximately 3500 µmol photons m$^{-2}$ s$^{-1}$ at the leaf surface. The HandyPEA measured the fluorescence intensity emitted in response to the actinic light. No measuring light was used and between pulses there was no light. To correct for differences in the fluorescence intensity due to differences in the actinic light intensity, the measured fluorescence values were divided by $F_0$ ( = $F_{20µs}$). Tobacco plants were taken from the growth chamber at the end of the night and were kept in near darkness for at least one hour before leaf clips were attached to a set of 10 leaves. These leaves were measured repetitively during the experiment with 10–15 min of darkness between measurements. Two types of measurements were carried out. To estimate the effective antenna size of PSII, leaves were illuminated with light intensities from 3500 down to 200 µmol photons m$^{-2}$ s$^{-1}$, starting at the highest light intensity. To characterize the reoxidation properties of PSII and the rest of the photosynthetic electron transport chain, two strong pulses of light (0.5 s, 3500 µmol photons m$^{-2}$ s$^{-1}$) were given to the leaves spaced $\Delta t$ apart, with $\Delta t$ between 0.1 and 200 s.

## Acknowledgements

The authors thank Benjamin Bailleul for help with ECS spectra determination and Judith Schaefers for help with chromatography. This work was supported by De Nederlandse Organisatie voor Wetenschappelijk Onderzoek (NWO), Earth and Life Sciences (ALW), through a Vici grant and by the European Research Council through the ERC Consolidator grant 281341 (ASAP) (to RC), a grant from the European Research Council (ERC) under the European Union's Horizon 2020 research, and innovation programme (ERC-ADG-2014; grant agreement 669982) to RB.

## Additional information

### Funding

| Funder | Grant reference number | Author |
|---|---|---|
| Nederlandse Organisatie voor Wetenschappelijk Onderzoek | Vici | Roberta Croce |
| H2020 European Research Council | ERC CON 281341 | Roberta Croce |

| H2020 European Research Council | ERC ADG 669982 | Ralph Bock |

The funders had no role in study design, data collection and interpretation, or the decision to submit the work for publication.

## Author contributions
Pengqi Xu, Data curation, Formal analysis, Investigation, Writing - original draft; Volha U Chukhutsina, Formal analysis, Investigation, Methodology, Writing - original draft; Wojciech J Nawrocki, Formal analysis, Investigation, Methodology; Gert Schansker, Formal analysis, Investigation, Methodology, Writing - review and editing; Ludwik W Bielczynski, Investigation; Yinghong Lu, Daniel Karcher, Resources; Ralph Bock, Conceptualization, Resources, Writing - review and editing; Roberta Croce, Conceptualization, Supervision, Funding acquisition, Writing - original draft, Writing - review and editing

## Author ORCIDs
Wojciech J Nawrocki (iD) http://orcid.org/0000-0001-5124-3000
Ralph Bock (iD) http://orcid.org/0000-0001-7502-6940
Roberta Croce (iD) https://orcid.org/0000-0003-3469-834X

## Decision letter and Author response
Decision letter https://doi.org/10.7554/eLife.58984.sa1
Author response https://doi.org/10.7554/eLife.58984.sa2

## Additional files

### Supplementary files
• Transparent reporting form

### Data availability
All data used for this study are included in the manuscript or in the supporting information.

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
