## [Decision Letter]

**Acceptance summary:**

As indicated by all the reviewers the work is very interesting and has broad implications for the importance of carotene and carotenoids in photosynthesis. The revised version addressed all the issues and nearly all the suggestions proposed by the reviewers. It is anticipated that this publication will inspire additional work that covers the remaining unknowns.

**Decision letter after peer review:**

Thank you for submitting your article "β-carotene is not essential for photosynthesis in higher plants" for consideration by *eLife*. Your article has been reviewed by three peer reviewers, including David M Kramer as the Reviewing Editor and Reviewer

#1, and the evaluation has been overseen by Christian Hardtke as the Senior Editor. The following individuals involved in review of your submission have agreed to reveal their identity: David Tiede (Reviewer #2); Alison Telfer (Reviewer #3).

The reviewers have discussed the reviews with one another and the Reviewing Editor has drafted this decision to help you prepare a revised submission.

The reviews were enthusiastic about the novelty of this work, and its potential implications for understanding the roles of carotenoids in photosynthesis. However, there was a consensus that the claims are too over-reaching. It was not demonstrated that the transgenic plants are not fully devoid of β-carotone and that other carotenoid species were also displaced. Further, while they may be able to grow under lab conditions but would not compete in the field. It is also not clear if there was clear substitution of Asta for other carotenoids, or if certain sites were left empty. Finally, there is an unresolved issue regarding the identification of the NPQ process that was observed in the Asta plants. Is it really equivalent to qE? This past issue could be addressed with some relatively easy experiments. However, considering the difficulty in performing more extensive work, the reviewers feel that a modified version of the paper might be acceptable, if it more realistically reflects the possible interpretations of the results, and eliminates overstatements, including reworking the title and Discussion. More specifically, the focus might be on the surprisingly small effects of substitution of specific carotenoids for others, rather than the essentiality of one specific case.

Reviewer #1:

This manuscript describes the characterization of a very interesting transgenic tobacco plant in which astaxanthin (asta) synthesis has been introduced. Asta accumulates to the extent that it outcompetes most if not all of the native carotenoids in photosynthetic complexes. The surprising finding is that the photosynthetic apparatus appears to function relatively normally, calling into question that textbook view that β-carotene is "essential" for photosynthesis. Overall, this conclusion is quite important for the field, and would encourage the authors to emphasize the danger of using data from knockout mutants to make claims of "essentiality". Their data clearly shows that photosynthesis can operate while β-carotone is largely replaced with (emphasis on these three words) asta. On the other hand, the current text also makes an over-reaching claim by stating (in the title and throughout the text) that β-carotone is not needed for photosynthesis. There are two important points that need to be clearly stated. First, they do not really show that β-carotone is not needed. They could do this by rescuing a mutant that lacks β-carotone by over-expressing asta, but those results are not shown. Second, the title is not accurate because they, rather than showing β-carotone is not needed, a difference compound can substitute for it. That this is possible is, nevertheless, both important and interesting. Although framing the story in this way makes a nice, snappy title, it very much distracts from what is, in fact, interesting and important in this work. The component on NPQ is quite interesting, but could be improved with some additional experiments or analysis.

"These data not only show that PSII is stable in the absence of β-carotene, but also demonstrate that the complexes are functional even when some of the carotenoid-binding sites are empty, as indicated by the higher Chl/car ratio in the Asta complexes compared to the WT complexes."

I am not convinced that this conclusion can be stated with certainty. If the Asta is more weakly bound some of it could have been lost during isolation, resulting in the apparently empty sites.

"Interestingly, the altered carotenoid composition has virtually no effect on the binding and spectroscopic properties of the Chls associated with the LHCs and the PSII core, as revealed by the high similarity of absorption (Figure 3—figure supplement 3A) and circular dichroism (Figure 3—figure supplement 3B) spectra of WT and mutant complexes in the Chl absorption regions…"

This seems to be overstated. There are differences, so it depends on what the definition of "virtually no effect" is. It would be better to tone it down a bit. There are so many cases where knocking out a gene has no apparent effect in the lab, but has strong effects in the field.

"Our data show that the difference in composition between WT and mutant complexes has only a small impact (5-10 °C) on their stability (Figure 3A)."

Similarly, I question whether 5-10 °C is "only a small impact" on stability. The WT would be able to live quite happily after a transient temperature spike, as experienced in the real world, but the asta plant would almost certainly die.

"Next, we investigated the effect of the absence of carotenes on the light-harvesting and trapping properties of the photosynthetic complexes in vivo by performing time-resolved fluorescence measurements on intact leaves (Figure 4 and Figure 4—figure supplement 1)."

Seems to suggest that there are no carotenoids in the leaves, which is obviously not true. This is apparent in many places in the text, and even more so in in the title, which seems to imply that carotenoids are not needed. In fact, they are not gone, only largely (with emphasis) replaced by Asta. What about changing the wording to something like, "…we investigated the effects of outcompeting carotenes with Ast…"?

"We verified that the mutant plants exhibit an ECS signal and we determined its action spectrum, which agreed with the prediction that astaxanthin is solely responsible for this in vivo Stark effect (Figure 5—figure supplement 1)." The data presented in Figure 5—figure supplement 1 is not an action spectrum. It is the light-induced difference spectrum. An action spectrum would be interesting, and would say something about the efficiency of light harvesting by and exciton transfer from Asta to the reaction centers.

"Indeed, comparison of the steady-state photochemical yields of PSII and PSI revealed that, at all light intensities, in both WT and mutant plants, the balance between PSII and PSI photochemistry is maintained (Figure 5A and Figure 5—figure supplement 3), meaning that the plants are able to compensate for the strong decrease in the PSII functional antenna size by decreasing the PSI/PSII ratio."

I disagree with the interpretation of these results at several levels. First, the calculation of NPQ used in the work is assumes that FM (relative fluorescence yield measured in the dark) occurs in the absence of any NPQ. In fact, what is indicated as NPQ is inaccurate because the plants were "pre-quenched" by the presence of Asta, so the NPQ is strongly underestimated. This is, of course, acknowledged in the text, "Note that the difference in NPQ level (1.8 in the WT vs. 0.3 in the mutant) can be fully ascribed to the presence of the static quencher in the mutant, which strongly reduces the maximal fluorescence in both dark (FM) and light (FM' ) states." It would be interesting to try the NPQ(t) parameter, which does not make this assumption.

Second, in a steady-state of linear electron flow, the apparent quantum efficiencies of PSI and PSII must, by definition, be equal, so the figure is not surprising. For example, plots of φII versus PAR are very similar in wild type and mutants deficient in NPQ, e.g. npq4, which lacks the qE response. What is different is that in the wild type, NPQ plays a stronger role in decreasing φII, whereas in npq4, it is accumulation of electrons on QA. Given the data in Figure 5B, I would expect that something similar occurs in the Asta plants, and it would be important to show this.

Third, the qP parameter is not is not a good linear indicator of the redox state of QA, nor is the so-called φI parameter are linear indicators of the quantum efficiency of PSI. The qL parameter might be a more linear indicator of QA redox state and plot of the qL versus the fraction of P700 in its oxidized state might be a better approach.

"It is likely that the high amount of PsbS in the mutant (Figure 2—figure supplement 1) can compensate for the lack of the xanthophyll cycle, or that astaxanthin can also be responsible for the dynamic quenching." It was shown by Li et al. (2009, Plant Cell 21, 1798-1812) that, in mutants lacking the xanthophyll cycle, lutein could restore qE. This reference provides some precedent for the observed persistence of rapid NPQ in line lacking Z.

The results on NPQ are not very thorough and some additional experiments are merited and, with minimal effort, make the paper stronger. In past work, several assays have been used to establish if an observed NPQ is qE, including addition of nigericin to inhibit lumen pH changes, elimination of VDE or ZE activity (e.g. mutation or addition of DTT), mutation of PsbS, or observation of the characteristic shifted spectrum associated with the qE process. It is probably not practical to ask for generation of PsbS/VDE/ZE mutants in the Asta background. However, it is relatively easy to perform the other experiments, and the outcomes would be very helpful in interpreting the results. For instance, Li et al., 2009, see above, found that the substitution of Lut for Z led to a shift in the qE-associated absorbance signal, showing that the onset of qE in the modified line still involved alteration of the spectral properties of a bound carotenoid.

Reviewer #2:

This is an excellent manuscript provides new information on the in vivo function of β-carotene on the assembly and function of PSII, PSI and LHC in higher plants. The manuscript is remarkable because it provides very clear demonstration that, contrary to widely accepted convention, β-carotene is not essential for the assembly and function of PSII and other light-harvesting complexes in tobacco. The results provide an important counterpoint the only other available mutant data from *Chlamydomonas*. As such, this manuscript will be of significant impact and a reference milestone. The manuscript is nicely placed in scientifically context, expertly documented with a broad suite of diagnostic spectroscopic tools, and generally very well-written.

Reviewer #3:

This is a very interesting paper challenging the belief that carotenes are essential for the stabilisation of PSII and PSI core complexes and hence essential for photosynthetic activity in oxygenic organisms. Mutant tobacco plants, totally lacking β-carotene and the normal xanthophylls, but making the xanthophyll, astaxanthin instead, can grow photosynthetically, all be it at a slower rate than the wild type. The mutant is characterised in this paper and it is shown that both photosystems and their respective light harvesting complexes bind astaxanthin. The paper reads well and is suitable for publication after dealing with the following points.

Note – I have numbered the pages from the title page throughout the review.

1) Abstract – last sentence. The text of the paper does not seem to me to make this point.

2) Figure 1 and Table 1: The figure shows a very orange/red plant but it is said to be 20 weeks old – whereas the table says that mature plants (Materials and methods – 20 weeks) have a very similar Chl/Car ratio to the WT and so I would guess they would be more green. Surely the Asta plant in Figure 1 is younger than 21 weeks? However, astaxanthin maybe be red shifted compared to β-carotene and the normal xanthophylls and so all plants are much redder than the WT? Clarify text.

3) A notable result in the paper is that the Asta mutant in all complexes binds astaxanthin but nearly half the sites are empty in both the PSI and PSII core complexes of Asta whereas the LHCs have only ~20% of the sites unfilled in Asta compared to WT (Table 2). The fact that the plants grow under lab conditions does not mean that the substitution of carotene for a xanthophyll does not have a drastic effect on photosynthetic activity. This should be discussed in more detail than the brief mention in the Results and Discussion.

4) Mention is made that PSI complexes from the mutant are more stable than PSII complexes (Results and Discussion and Figure 3) but there is no mention that PSII is intrinsically less stable than PSI. It is well known that in PSII, carotenoids cannot protect by quenching singlet oxygen formed via P680 triplet because of the very high oxidising potential of PSII.

5) Results and Discussion and Table 3 re ratio PSII/PSI: The text is very confusing and not consistent. The data does not show that this ratio is the same whether measured by ECS or protein levels. ECS is 2.6 times higher while by protein levels the PSI/PSII ratio it is said to be 6 times lower (Results and Discussion). Discuss this difference and use the same ratio PSII/PSI when reporting data and discussing in text.

---

## [Author Response]

Reviewer #1:This manuscript describes the characterization of a very interesting transgenic tobacco plant in which astaxanthin (asta) synthesis has been introduced. Asta accumulates to the extent that it outcompetes most if not all of the native carotenoids in photosynthetic complexes. The surprising finding is that the photosynthetic apparatus appears to function relatively normally, calling into question that textbook view that β-carotene is "essential" for photosynthesis. Overall, this conclusion is quite important for the field, and would encourage the authors to emphasize the danger of using data from knockout mutants to make claims of "essentiality". Their data clearly shows that photosynthesis can operate while β-carotone is largely replaced with (emphasis on these three words) asta. On the other hand, the current text also makes an over-reaching claim by stating (in the title and throughout the text) that β-carotone is not needed for photosynthesis. There are two important points that need to be clearly stated. First, they do not really show that β-carotone is not needed. They could do this by rescuing a mutant that lacks β-carotone by over-expressing asta, but those results are not shown. Second, the title is not accurate because they, rather than showing β-carotone is not needed, a difference compound can substitute for it. That this is possible is, nevertheless, both important and interesting. Although framing the story in this way makes a nice, snappy title, it very much distracts from what is, in fact, interesting and important in this work. The component on NPQ is quite interesting, but could be improved with some additional experiments or analysis.

We are glad the reviewer finds our work important for the field. We have changed the title to “Photosynthesis without β-carotene”. We think this title captures the essence of our work. We agree that, unless we can demonstrate that there is not even only one molecule of β-carotene per plant, we cannot completely exclude its role in photosynthesis. However, the canonical role of β-carotene is linked to its association with PSI and PSII. What we can safely conclude is that, in our plants, the vast majority of the complexes does not contain any β-carotene molecules.

"These data not only show that PSII is stable in the absence of β-carotene, but also demonstrate that the complexes are functional even when some of the carotenoid-binding sites are empty, as indicated by the higher Chl/car ratio in the Asta complexes compared to the WT complexes."I am not convinced that this conclusion can be stated with certainty. If the Asta is more weakly bound some of it could have been lost during isolation, resulting in the apparently empty sites.

We have now added a sentence to explain that we cannot exclude that astaxanthin is lost upon purification and we have also reworded this sentence to make it clearer that it refers to the isolated complex in which some of the binding sites are “empty” in the sense that they are not occupied by carotenoids. This is now also explicitly mentioned.

"Interestingly, the altered carotenoid composition has virtually no effect on the binding and spectroscopic properties of the Chls associated with the LHCs and the PSII core, as revealed by the high similarity of absorption (Figure 3—figure supplement 3A) and circular dichroism (Figure 3—figure supplement 3B) spectra of WT and mutant complexes in the Chl absorption regions…"This seems to be overstated. There are differences, so it depends on what the definition of "virtually no effect" is. It would be better to tone it down a bit. There are so many cases where knocking out a gene has no apparent effect in the lab, but has strong effects in the field.

We have reformulated the sentence and we are referring the reader to the Figure 3—figure supplement 3 where we provide a more detailed explanation of the spectra.

"Our data show that the difference in composition between WT and mutant complexes has only a small impact (5-10 °C) on their stability (Figure 3A)."Similarly, I question whether 5-10 °C is "only a small impact" on stability. The WT would be able to live quite happily after a transient temperature spike, as experienced in the real world, but the asta plant would almost certainly die.

We have rephrased this sentence to make it clearer that we measure isolated complexes and that the results of these measurements are only valid for the isolated complexes.

"Next, we investigated the effect of the absence of carotenes on the light-harvesting and trapping properties of the photosynthetic complexes in vivo by performing time-resolved fluorescence measurements on intact leaves (Figure 4 and Figure 4—figure supplement 1)."Seems to suggest that there are no carotenoids in the leaves, which is obviously not true. This is apparent in many places in the text, and even more so in in the title, which seems to imply that carotenoids are not needed. In fact, they are not gone, only largely (with emphasis) replaced by Asta. What about changing the wording to something like, "…we investigated the effects of outcompeting carotenes with Ast…"?

We have changed the sentence into “Next, we investigated the effect of the substitution of carotenes with astaxanthin”…. We also changed a similar sentence at the end of the Discussion. We are mentioning throughout the manuscript that our complexes contain astaxanthin and the definition of “carotenes” is already introduced in the Abstract. The title mentioned explicitly β-carotene, and this was done to avoid creating confusion to readers who may not be familiar with the difference between “carotenoids” and “carotenes”.

"We verified that the mutant plants exhibit an ECS signal and we determined its action spectrum, which agreed with the prediction that astaxanthin is solely responsible for this in vivo Stark effect (Figure 5—figure supplement 1)." The data presented in Figure 5—figure supplement 1 is not an action spectrum. It is the light-induced difference spectrum. An action spectrum would be interesting, and would say something about the efficiency of light harvesting by and exciton transfer from Asta to the reaction centers.

We have reformulated the sentence as suggested.

"Indeed, comparison of the steady-state photochemical yields of PSII and PSI revealed that, at all light intensities, in both WT and mutant plants, the balance between PSII and PSI photochemistry is maintained (Figure 5A and Figure 5—figure supplement 3), meaning that the plants are able to compensate for the strong decrease in the PSII functional antenna size by decreasing the PSI/PSII ratio."I disagree with the interpretation of these results at several levels. First, the calculation of NPQ used in the work is assumes that FM (relative fluorescence yield measured in the dark) occurs in the absence of any NPQ. In fact, what is indicated as NPQ is inaccurate because the plants were "pre-quenched" by the presence of Asta, so the NPQ is strongly underestimated. This is, of course, acknowledged in the text, "Note that the difference in NPQ level (1.8 in the WT vs. 0.3 in the mutant) can be fully ascribed to the presence of the static quencher in the mutant, which strongly reduces the maximal fluorescence in both dark (FM) and light (FM' ) states." It would be interesting to try the NPQ(t) parameter, which does not make this assumption.

We have now used the NPQ(t) parameter, which shows that the level is even higher in the mutant than in the WT, in agreement with the higher amount of PsbS in the former. The data are shown in Figure 5—figure supplement 7.

Second, in a steady-state of linear electron flow, the apparent quantum efficiencies of PSI and PSII must, by definition, be equal, so the figure is not surprising. For example, plots of φII versus PAR are very similar in wild type and mutants deficient in NPQ, e.g. npq4, which lacks the qE response. What is different is that in the wild type, NPQ plays a stronger role in decreasing φII, whereas in npq4, it is accumulation of electrons on QA. Given the data in Figure 5B, I would expect that something similar occurs in the Asta plants, and it would be important to show this.

We agree with the reviewer that the apparent quantum yields in steady-state (and in the linear electron flow regime) will respond in the same way to changes in the light intensity, as will the ETRs of PSII and PSI. This is demonstrated in our work: as the reviewer points out, global parameters relating to PSII (qL) and PSI (phi PSI) both show virtually identical light dependence in asta- and WT plants. Crucially, the similarity of NPQ(t) parameter amplitude and kinetics between asta and WT (Figure 5—figure supplement 7) shows that the presence of static quenching can fully explain the difference in apparent NPQ. In consequence, given that the asta plants exhibit an NPQ(t) of ~2.3 already in darkness, we share reviewer’s opinion that in steady-state, all other parameters unchanged (like in the *npq4* example used), the static quenching would be by far the biggest reason for a change/decrease in the PSII ETR light intensity dependence. The reviewer observes that in the npq4 mutant more Q_A_ becomes reduced. In Figure 5—figure supplement 3 it is shown that in asta-plants this is not the case, in agreement with the strong static and NPQ quenching in these plants.

Instead, we substantiate by measuring the RC ratio- and antenna sizes *per PS* that the optimisation of photosynthesis in the mutant is achieved through the regulation of the relative stoichiometries of the photosystems. Unfortunately, this makes the situation significantly more complex than for WT-*npq4*. We think that quantification of the partitioning of the effect of NPQ and b6f turnover rate for this difference would be difficult due to the superposition of static quenching, changes in PS stoichiometries, altered antenna sizes etc.

Nonetheless, in our opinion such small differences in the qL/phi PSI/light intensity plots rather highlight how remarkably well the mutant adapts to its genetic engineering and accounts for the necessity to maintain PSII and PSI ETRs even, as well as the b6f ETR capacity such that it exhibits pH-dependence (and thus the extent of photosynthetic control) similar to this in the WT.

Third, the qP parameter is not is not a good linear indicator of the redox state of QA, nor is the so-called φI parameter are linear indicators of the quantum efficiency of PSI. The qL parameter might be a more linear indicator of QA redox state and plot of the qL versus the fraction of P700 in its oxidized state might be a better approach.

We have calculated qL and we now show it in Figure 5. We still show qP in the Figure 5—figure supplement 4, to highlight the similar dependence of the data to fPSI or PAR. The fractions of open/closed PSII centres following a pulse were calculated in Figure 5—figure supplement 5.

"It is likely that the high amount of PsbS in the mutant (Figure 2—figure supplement 1) can compensate for the lack of the xanthophyll cycle, or that astaxanthin can also be responsible for the dynamic quenching." It was shown by Li et al. (2009, Plant Cell 21, 1798-1812) that, in mutants lacking the xanthophyll cycle, lutein could restore qE. This reference provides some precedent for the observed persistence of rapid NPQ in line lacking Z.

Both lutein and zeaxanthin are explicitly mentioned in the Abstract and in the Results and Discussion. We have now added the reference to Li et al., 2009.

The results on NPQ are not very thorough and some additional experiments are merited and, with minimal effort, make the paper stronger. In past work, several assays have been used to establish if an observed NPQ is qE, including addition of nigericin to inhibit lumen pH changes, elimination of VDE or ZE activity (e.g. mutation or addition of DTT), mutation of PsbS, or observation of the characteristic shifted spectrum associated with the qE process. It is probably not practical to ask for generation of PsbS/VDE/ZE mutants in the Asta background. However, it is relatively easy to perform the other experiments, and the outcomes would be very helpful in interpreting the results. For instance, Li et al., 2009, see above, found that the substitution of Lut for Z led to a shift in the qE-associated absorbance signal, showing that the onset of qE in the modified line still involved alteration of the spectral properties of a bound carotenoid.

This is unfortunately not possible, because due to Covid-19, we had to empty all our growth chambers and it will take months before we will have plants at the right stage for these measurements. Nonetheless, we are confident that we observe qE, because the relaxation of NPQ is very rapid, with identical kinetics to the WT (see inset in Figure 5B) despite the presence of static quenching. We are also reporting now the NPQ(t) parameter, which shows that the induced quenching is substantial.

Reviewer #3:This is a very interesting paper challenging the belief that carotenes are essential for the stabilisation of PSII and PSI core complexes and hence essential for photosynthetic activity in oxygenic organisms. Mutant tobacco plants, totally lacking β-carotene and the normal xanthophylls, but making the xanthophyll, astaxanthin instead, can grow photosynthetically, all be it at a slower rate than the wild type. The mutant is characterised in this paper and it is shown that both photosystems and their respective light harvesting complexes bind astaxanthin. The paper reads well and is suitable for publication after dealing with the following points.Note – I have numbered the pages from the title page throughout the review.1) Abstract – last sentence. The text of the paper does not seem to me to make this point.

We refer here to the large change in PSI/PSII ratio between WT and the mutant, which is discussed in the last part of the manuscript. We have added a sentence to make the link to the Abstract clearer.

2) Figure 1 and Table 1: The figure shows a very orange/red plant but it is said to be 20 weeks old – whereas the table says that mature plants (Materials and methods – 20 weeks) have a very similar Chl/Car ratio to the WT and so I would guess they would be more green. Surely the Asta plant in Figure 1 is younger than 21 weeks? However, astaxanthin maybe be red shifted compared to β-carotene and the normal xanthophylls and so all plants are much redder than the WT? Clarify text.

We are referring to the age of the leaves. The older leaves are greener than the young ones. This is now mentioned in the legend. We have also added Figure 1—figure supplement 1 with photographs of plants of different ages.

3) A notable result in the paper is that the Asta mutant in all complexes binds astaxanthin but nearly half the sites are empty in both the PSI and PSII core complexes of Asta whereas the LHCs have only ~20% of the sites unfilled in Asta compared to WT (Table 2). The fact that the plants grow under lab conditions does not mean that the substitution of carotene for a xanthophyll does not have a drastic effect on photosynthetic activity. This should be discussed in more detail than the brief mention in the Results and Discussion.

These results are based on purified complexes, we have now better clarified this point and mentioned that we cannot exclude that some of the astaxanthin molecules are lost during purification. We have also added a sentence to the Introduction, mentioning that these plants can grow without β-carotone at least in laboratory conditions.

4) Mention is made that PSI complexes from the mutant are more stable than PSII complexes (Results and Discussion and Figure 3) but there is no mention that PSII is intrinsically less stable than PSI. It is well known that in PSII, carotenoids cannot protect by quenching singlet oxygen formed via P680 triplet because of the very high oxidising potential of PSII.

This was indeed the point that we wanted to make: that PSI with few carotenoids is still better protected than PSII. We have now emphasized this point as suggested.

5) Results and Discussion and Table 3 re ratio PSII/PSI: The text is very confusing and not consistent. The data does not show that this ratio is the same whether measured by ECS or protein levels. ECS is 2.6 times higher while by protein levels the PSI/PSII ratio it is said to be 6 times lower (Results and Discussion). Discuss this difference and use the same ratio PSII/PSI when reporting data and discussing in text.

Thank you. We have now corrected it and discussed the difference.